# Simulation-Free Differential Dynamics through Neural Conservation Laws

## Abstract

We present a novel simulation-free framework for training continuous-time diffusion processes over very general objective functions. Existing methods typically involve either prescribing the optimal diffusion process—which only works for heavily restricted problem formulations—or require expensive simulation to numerically obtain the time-dependent densities and sample from the diffusion process. In contrast, we propose a coupled parameterization which jointly models a time-dependent density function, or probability path, and the dynamics of a diffusion process that generates this probability path. To accomplish this, our approach directly bakes in the Fokker-Planck equation and density function requirements as hard constraints, by extending and greatly simplifying the construction of Neural Conservation Laws. This enables simulation-free training for a large variety of problem formulations, from data-driven objectives as in generative modeling and dynamical optimal transport, to optimality-based objectives as in stochastic optimal control, with straightforward extensions to mean-field objectives due to the ease of accessing exact density functions. We validate our method in a diverse range of application domains from modeling spatio-temporal events to learning optimal dynamics from population data.

## 1 Introduction

Diffusion models have been widely adopted due to their ease of use and competitive performance in generative modeling Ho et al. (2020); Ma et al. (2024); Chen & Lipman (2024), by learning a diffusion process that interpolates between a data distribution and a Gaussian noise distribution Song et al. (2021); Albergo et al. (2023); Lipman et al. (2023). However, their construction is heavily restrictive and only results in a simulation-free training algorithm for this simplest case. Recent works have adapted these ideas to train diffusion processes over more general objective functions, such as solving optimal transport or generalized Schrödinger bridge problems, but these methods all require simulating from the learned diffusion process to some varying degrees, and are generally more restrictive than simulation-based training approaches Liu et al. (2024).

We consider training diffusion processes over general objective functions[1]

$$\min_{\rho, u} \quad \int_0^1 L(\rho_t, u_t)dt + F(\rho_0, \rho_1) \tag{1}$$

$$\text{s.t.} \quad \partial_t \rho_t = -\nabla \cdot (u_t \rho_t) + \tfrac{1}{2}g_t^2 \Delta \rho_t \tag{2}$$

$$\rho_t \geq 0, \quad \int_{\mathbb{R}^D} \rho_t(x)dx = 1 \quad \forall t \in [0,1] \tag{3}$$

where $u_t(x) : \mathbb{R}^{1+D} \to \mathbb{R}^D$ and $\rho_t(x) : \mathbb{R}^{1+D} \to \mathbb{R}^+$ are the time-dependent velocity field and probability density function to be learned, and $g_t$ is a state-independent diffusion coefficient that is given as part of the problem. The functionals $L$ and $F$ can be quite general, including cases such as generative modeling from data observations, Schrödinger bridge problems, and mean-field control—we provide concrete examples in Section 4. The constraints in eq. (3) ensure the density

---

[1]We denote $\partial_t \rho_t = \frac{\partial \rho_t}{\partial t}$, $\nabla \cdot (u_t \rho_t) = \sum_{d=1}^D \frac{\partial (u_t \rho_t)}{\partial x_d}$, and $\Delta \rho_t = \sum_{d=1}^D \frac{\partial^2 \rho_t}{\partial x_d^2}$.

function is properly normalized, while the constraint in eq. (2)—the Fokker-Planck equation—implies that the diffusion process modeled by the stochastic differential equation (SDE)

$$dX_t = u_t(X_t)dt + g_t dW_t \tag{4}$$

transports particles in accordance with the marginal densities, *i.e.* $X_t \sim \rho_t$.

Typical approaches will only directly parameterize $u_t$, the time-evolution of the particles, whereas $\rho_t$, the time-evolution of probability density function, is either unobtainable or only estimated through expensive numerical procedures (Chen et al., 2019b; Kobyzev et al., 2021). As such, in order to sample from $\rho_t$, typically one transports particles starting from the initial time to time $t$, *simulating* the diffusion process in eq. (4).

In this work, we propose a novel parameterization of diffusion processes where we parameterize not only the dynamics $u_t$ but also the density $\rho_t$ in an explicit form, then we direct impose the Fokker-Planck equation (2) as a hard constraint on the model in order to couple these two quantities. To do so, we build upon ideas from Neural Conservation Laws (NCL; Richter-Powell et al. (2022)) for imposing the continuity equation. We propose a reformulation of the NCL framework and significantly improve upon its prior construction; unlike prior work, we additionally include the *density constraints* (3) into the model, enabling maximum likelihood training. We also find that the naïve construction introduces what we call a *spurious flux phenomenon* which renders the velocity field unusable. We propose removing this phenomenon through the introduction of a carefully designed divergence-free component into the dynamics model that leaves the density invariant. In summary, our work introduces the following contributions:

- Improved analysis of the Neural Conservation Laws construction, generalizing to diffusion processes and additionally imposing the density constraints (3). Compared to the original formulation, we can now train with the maximum likelihood objective.

- We discuss how the naïve construction leads to a spurious flux phenomenon, where the flux and velocity field do not vanish even as $x$ diverges. We mitigate this by introducing carefully chosen divergence-free components to the flux that leaves the density invariant.

- We show that our method achieves state-of-the-art on a variety of spatio-temporal generative modeling data sets and on learning transport maps in cellular dynamics.

- To the best of our knowledge, we are the first method to be able to train a diffusion process with general objective functions—such as regularizing towards optimal transport, or with additional state costs, including mean-field cost functions—completely simulation-free, whereas existing methods require varying degrees of simulation.

## 2 RELATED WORK

Markov processes described by ordinary and stochastic differential equations have been used across many application domains (Rubanova et al., 2019; Karniadakis et al., 2021; Cuomo et al., 2022; Wang et al., 2023), with the most general problem settings involving simulation-based methods. This refers to training neural differential equations of various kinds by simulating their trajectories and differentiating through the objective function. While some works have solved the memory issue with dfferentating through simulations (Chen et al., 2020; Li et al., 2020; Chen et al., 2021), it remains problematic to apply these at scale due to the computational cost of simulation. Furthermore, many probabilistic modeling applications (Grathwohl et al., 2018; Chen et al., 2019a; Koshizuka & Sato, 2023) require the computation of the likelihood for maximum likelihood training, which can be more expensive than simulating trajectories.

This is where Neural Conservation Laws (NCL; Richter-Powell et al. (2022)) come in, which is a modeling paradigm where the law of conservation such as eq. (2) is directly enforced as a hard constraint. This allows optimization of the kind in eq. (1) to be mapped an unconstrained problem in the parameter space of an NCL model. However, while the original NCL model (Richter-Powell et al., 2022) was able to bake in the constraint in eq. (2), they did not provide a scalable way to bake in the density constraints in eq. (3) which is key for enabling maximum likelihood training.

A highly-scalable approach is the framework of diffusion models (Ho et al., 2020; Song et al., 2021), Flow Matching (Lipman et al., 2023), and stochastic interpolants (Albergo et al., 2023). However,

these methods can only solve a restricted set of problems, ones where samples from the optimal $\rho_0$ and $\rho_1$ are provided for training. They cannot handle the general problem setup of eq. (1) but instead directly prescribe the optimal solution which is then learned by a regression problem.

## 3 METHOD

We describe a novel framework which directly parameterizes both a velocity field $u_t$ and a density $\rho_t$ that always satisfies the Fokker-Planck constraint in eq. (2) and density constraints in eq. (3). Our method is built on top of ideas introduced in Neural Conservation Laws (NCL; Richter-Powell et al. (2022)) through the use of differential forms, but we take an alternative construction while providing step-by-step derivations. We then discuss how likelihood-based generative models can fit within our framework. The naïve construction, however, leads to a problem we call the *spurious flux phenomenon* (Section 3.4) which we resolve by introducing a divergence-free component (Section 3.5).

### 3.1 NEURAL CONSERVATION LAWS

In order to satisfy the Fokker-Planck constraint in eq. (2), we make use of a coupled parameterization of both a *probability path* $\rho_t$, *i.e.* a time-dependent density function, and a *flux* $j_t(x) : \mathbb{R}^{D+1} \to \mathbb{R}^D$ that is designed, by construction, to always satisfy the continuity equation,

$$\partial_t \rho_t + \nabla \cdot j_t = 0. \tag{5}$$

This equation imposes the condition that the total probability in a system must be conserved, and that instantaneous changes in the probability can only be attributed to the local movement of particles following a continuous flow characterized together by $j_t$ and $\rho_t$.

We directly impose the continuity equation into the model as a hard constraint. This idea was previously explored in Neural Conservation Laws (NCL; Richter-Powell et al. (2022)); however, its reliance on differential forms makes it difficult to extend, and they were not able to satisfy the density constraints in eq. (3). Instead, we propose a simplified alternative construction and will derive the core building blocks of NCL that are necessary for our approach following only basic principles.

To model eq. (5), we introduce two vector fields $a_t^\theta(x) : \mathbb{R}^{1+D} \to \mathbb{R}^D$ and $b_t^\theta(x) : \mathbb{R}^{1+D} \to \mathbb{R}^D$ with parameters $\theta$, and set

$$\rho_t = \nabla \cdot a_t^\theta, \tag{6}$$

$$j_t = -\partial_t a_t^\theta + b_t^\theta. \tag{7}$$

With this choice we have:

**Lemma 1.** *Let $\rho_t$ and $j_t$ be given by eq. (6) and eq. (7), respectively. Then the continuity eq. (5) holds iff $b_t$ is divergence-free, i.e. $\nabla \cdot b_t^\theta = 0$.*

*Proof.* Plugging eq. (6) and eq. (7) into the left hand side of eq. (5),

$$\partial_t \rho_t + \nabla \cdot j_t = \partial_t \nabla \cdot a_t^\theta - \nabla \cdot (\partial_t a_t^\theta + b_t^\theta) = -\nabla \cdot b_t^\theta. \tag{8}$$

Therefore eq. (5) holds iff $\nabla \cdot b_t^\theta = 0$, which verifies the claim. $\square$

Notice that $\rho_t$ depends only on $a_t^\theta$, while $j_t$ is affected by both $a_t^\theta$ and $b_t^\theta$. The extra degrees of freedom coming from $b_t^\theta$ will be important in order to resolve what we call the *spurious flux phenomenon* in Section 3.4, and furthermore, it provides the needed flexibility in order to learn optimal solutions of $u_t$ while leaving $\rho_t$ invariant.

### 3.2 CONVERSION TO DIFFERENTIAL DYNAMICS

In order to obtain the dynamics directly, we need to convert the continuity equation into the Fokker-Planck equation. Fortunately, the density and flux provide sufficient information in order to perform this conversion. Any flux $j_t$ that satisfies the continuity equation in eq. (5) can be converted to a $u_t$ that satisfies eq. (2) using the following identity:

$$u_t = \frac{j_t}{\rho_t} + \tfrac{1}{2} g_t^2 \nabla \log \rho_t. \tag{9}$$

This can be verified by plugging eq. (9) into eq. (2).

$$\partial_t \rho_t = -\nabla \cdot \left( \rho_t \left( \frac{j_t}{\rho_t} + \frac{1}{2} g_t^2 \nabla \log \rho_t \right) \right) + \frac{1}{2} g_t^2 \Delta \rho_t(x)$$
$$= -\nabla \cdot \left( j_t + \frac{1}{2} g_t^2 \nabla \rho_t \right) + \frac{1}{2} g_t^2 \Delta \rho_t(x) = -\nabla \cdot j_t \qquad (10)$$

Thus, by parameterizing a single vector field $a_t^\theta$, we can model both a density $\rho_t$ and a velocity field $u_t$ that satisfies the constraint in eq. (2). This allows us to turn the constrained optimization problem in eq. (1) into an unconstrained optimization in the parameters $\theta$. Furthermore, as we are given direct access to $\rho_t$, we do not need to solve the Fokker-Planck equation (eq. (2)) from $u_t$, typically requiring an extremely expensive procedure. This enables a new paradigm of simulation-free methods for training diffusion models over general objective functions.

## 3.3 Designing $a_t^\theta$ through likelihood-based models

In order to model valid probability density functions, we must also satisfy the density constraints in eq. (3). In addition, we wish to design our choice of $a_t^\theta$ such that (i) $\rho_t$ can be exactly sampled from at any time value $t$, (ii) computation of $\rho_t$ incurs minimal computational cost, and (iii) the model is flexible enough for practical applications. We will show that autoregressive likelihood-based models can nicely fit within our framework and satisfies all of the above desirables.

Consider a time-dependent autoregressive probabilistic model which decomposes the joint distribution over all $D$ variables given the natrual ordering,

$$\rho_t(x) = \prod_{i=1}^D f_t^\theta(x_i | x_{1:i-1}), \qquad (11)$$

and denote by $F_t^\theta(x_i | x_{1:i-1}) = \int_{-\infty}^{x_i} f_t^\theta(y | x_{1:i-1}) dy$ the associated cumulative probability distributions (CDF).

Letting $[a_t^\theta]_i$ denote the $i$-th coordinate of $a_t^\theta$, define

$$[a_t^\theta]_i(x) = \begin{cases} F_t^\theta(x_i | x_{1:i-1}) \prod_{j=1, j\neq i}^D f_t^\theta(x_j | x_{1:j-1}), & i = D \\ 0, & \text{otherwise} \end{cases} \qquad (12)$$

This allows us to model the density as $\rho_t(x) = \nabla \cdot a_t^\theta(x) = \prod_{i=1}^D f_t^\theta(x_i | x_{i-1})$, which we will refer to as the *autoregressive model*.

Alternatively, we can consider a factorized model, where $x_i$ do not depend on other variables:

$$f_t^\theta(x_i | x_{1:i-1}) = f_t^\theta(x_i), \qquad F_t^\theta(x_i | x_{1:i-1}) = F_t^\theta(x_i), \qquad (13)$$

which we will refer to as the *factorized model*.

**Choice of $F_t^\theta$ as mixture of logistics.** While one may directly parameterize the functions $F_t^\theta$ using monotonic neural networks (Sill, 1997; Daniels & Velikova, 2010), resulting in a universal density approximator, we decide to use a simpler construction using mixture of logistic distributions. Mixture of logistics has been a common choice among likelihood-based generative modeling frameworks, from normalizing flows (Kingma et al., 2016; Ho et al., 2019) to autoregressive models (Salimans et al., 2017). Similarly, mixture of logistics is sufficient flexible for our use cases, as we only need to model per-coordinate conditional distributions. For our autoregressive model, we use a mixture of logistics to describes the CDF as

$$F_t^\theta(x_i | x_{1:i-1}) = \sum_{l=1}^L \alpha_l^\theta(x_{1:i-1}, t) \left[ \sigma \left( s_l^\theta(x_{1:i-1}, t) \left( x_i - \mu_l^\theta(x_{1:i-1}, t) \right) \right) \right], \qquad (14)$$

where $\sigma(x) = 1/(1 + \exp(-x))$ is the sigmoid function. Here, $\mu_l^\theta(x_{1:i-1}, t)$ and $s_l^\theta(x_{1:i-1}, t)$ correspond to the mean and inverse scale of a logistic distribution, respectively, while $\alpha_l^\theta(x_{1:i-1}, t)$ are mixture weights. All functions are parameterized using autoregressive neural networks. These correspond to probability density functions

$$f_t^\theta(x_i | x_{1:i-1}) = \sum_{l=1}^L \alpha_l^\theta(x_{1:i-1}, t) \left[ s_l^\theta(x_{1:i-1}, t) \sigma \left( z_l^\theta(x_i, x_{1:i-1}, t) \right) \sigma \left( -z_l(x_i, x_{1:i-1}, t) \right) \right], \qquad (15)$$

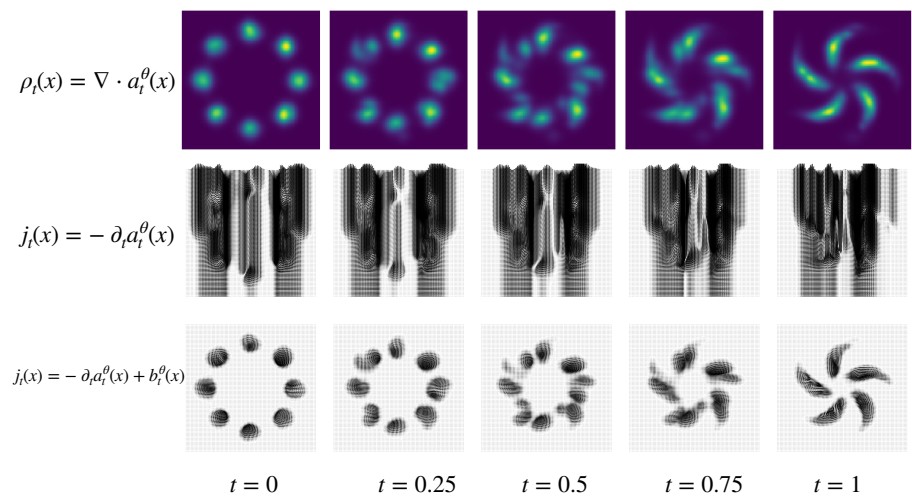

$$\rho_t(x) = \nabla \cdot a_t^\theta(x)$$

$$j_t(x) = -\partial_t a_t^\theta(x)$$

$$j_t(x) = -\partial_t a_t^\theta(x) + b_t^\theta(x)$$

| $t = 0$ | $t = 0.25$ | $t = 0.5$ | $t = 0.75$ | $t = 1$ |

Figure 1: Illustration of the *spurious flux phenomenon* and its removal with a divergence-free vector field $b_t^\theta$. (*top*) The trained marginal distributions in 2D. (*middle*) The flux field $j_t = -\partial_t a_t^\theta$ without any flux cancellations, where we see there are spurious fluxes. (*bottom*) The flux field $j_t = -\partial_t a_t^\theta + b_t^\theta$ with $b_t^\theta$ defined in Section 3.5, and we now see that the flux field vanishes properly.

where

$$z_l^\theta(x_i, x_{1:i-1}, t) = s_l^\theta(x_{1:i-1}, t)\left(x_i - \mu_l^\theta(x_{1:i-1}, t)\right). \tag{16}$$

As for the factorized model, since we remove all the dependecies of the CDF on the prior coordinates, we therefore can describe $F_t^\theta(x_i)$ as

$$F_t^\theta(x_i) = \sum_{l=1}^{L} \alpha_l^\theta(t)\left[\sigma\left(s_l^\theta(t)\left(x_i - \mu_l^\theta(t)\right)\right)\right], \tag{17}$$

where the mean, inverse scale, and the mixture weights are functions depend on $t$ only.

While these constructions for the factorized model and the autoregressive lead to a proper density, the keen reader may notice that the flux constructed from eq. (7) using this $a_t^\theta$ is problematic as the flux will be exactly zero in all but one coordinate. This is not the core of the problem but rather a manifestation of the spurious flux phenonmenon, which we will describe in Section 3.4. We will later go in depth on how to construct a proper flux by making use of the extra degree of freedom we have in designing $b_t^\theta$ later in Section 3.5.

## 3.4 THE SPURIOUS FLUX PHENOMENON

The choice of $a_t^\theta$ above guarantees that $\rho_t$ is positive and normalizes to exactly one. However, using only this $a_t^\theta$ and setting $b_t^\theta = 0$ in eq. (7) turns out to be problematic for the flux $j_t$. Indeed, given any box-shaped region $\mathcal{X} = [-L, L]^D$ with boundary denoted $\partial\mathcal{X}$ and normal vector $\hat{n}(x)$, we can use the divergence theorem to obtain

$$\int_\mathcal{X} \rho_t(x)dx = \int_\mathcal{X} \nabla \cdot a_t^\theta(x)dx = \int_{\partial\mathcal{X}} \hat{n}(x) \cdot a_t^\theta(x)dS(x) > 0. \tag{18}$$

where $dS(x)$ is the surface measure on $\partial\mathcal{X}$. This quantity is nonzero no matter how large $L$ is, and approaches one as $L \to \infty$ since $\int_{\mathbb{R}^D} \rho_t(x)dx = 1$.

This implies that $a_t^\theta$ is necessarily nonzero somewhere even outside of the support of $\rho_t$. Therefore, if we set $b_t^\theta = 0$ so that $j_t = -\partial_t a_t^\theta$ by eq. (7), since $a_t^\theta$ is not constant in $t$ in general, the flux does not decay to zero even outside the support of $\rho_t$, even though $\rho_t$ goes to zero. This is problematic for two reasons: (i) we take $j_t/\rho_t$ in order to construct $u_t$, which will diverge, and (ii) because the divergence theorem holds for *any* region, this introduces problematic behavior even at finite $x$, as can be seen in Figure 1.

We define this behavior where the flux is nonzero even as $x$ diverges as the *spurious flux phenomenon*. While this phenomenon exists generally for any construction that does not enforce $\lim_{x\to\infty} \partial_t a_t^\theta = 0$, we can gain a more concrete understanding of the spurious flux by considering the autoregressive construction of $a_t^\theta$ as in eq. (12), which we formalize in the following result:

**Lemma 2.** *Let* $\rho_t(x) = \nabla \cdot a_t^\theta(x) = \prod_{i=1}^{D} f_t^\theta(x_i)$ *and* $j_t(x) = -\partial_t a_t^\theta(x)$ *with* $a_t^\theta(x)$ *given by eq. (12). Then* $\lim_{x_i \to +\infty} |j_t(x)|^2 \neq 0$.

*Proof.* If $j_t(x) = -\partial_t a_t^\theta(x)$ with $a_t^\theta(x)$ given by eq. (12), the $i$-th coordinate of the flux is

$$[j_t]_i(x) = -\partial_t F_t^\theta(x_i|x_{1:i-1}) \prod_{i=1}^{D-1} f_t^\theta(x_i|x_{1:i-1}) - F_t^\theta(x_i|x_{1:i-1})\partial_t \left(\prod_{i=1}^{D-1} f_t^\theta(x_i|x_{1:i-1})\right).$$
(19)

Since $\lim_{x_i \to \infty} F_t(i) = 1$ and therefore $\lim_{x_i \to \infty} \partial_t F_t(i) = 0$, we deduce

$$\lim_{x_i \to \infty} [j_t]_i(x) = -\partial_t \left(\prod_{i=1}^{D-1} f_t^\theta(x_i|x_{1:i-1})\right) \neq 0$$
(20)

and the claim of the lemma follows. $\qquad\square$

This understanding of the spurious flux allow us to combat this phenomenon by cancelling out the problematic flux terms using the extra degree of freedom offered by the divergence-free field $b_t^\theta$ in the construction of eq. (7), as discussed next.

### 3.5 DESIGNING $b_t^\theta$ TO COMBAT THE SPURIOUS FLUX PHENOMENON

In order to remove the spurious flux in eq. (20), we must cancel it out with a term that has the same limiting behavior. To this end, we propose adding the quantity $\sigma(x_D)\partial_t \left(\prod_{i=1}^{D-1} f_t^\theta(x_i|x_{1:i-1})\right)$ to the $i$-th coordinate of the flux, where $\sigma$ is the sigmoid function, or more generally, a function/neural network approaches 1 as its input goes to infinity. By adding this *cancellation* term to the spurious flux from eq. (20), we exactly remove the limiting behavior:

$$\lim_{x_i \to \infty} -\partial_t \left(F_t^\theta(x_i|x_{1:i-1}) \prod_{i=1}^{D-1} f_t^\theta(x_i|x_{1:i-1})\right) + \sigma(x_i|x_{1:i-1})\partial_t \left(\prod_{i=1}^{D-1} f_t^\theta(x_i|x_{1:i-1})\right) = 0.$$
(21)

However, we must construct $b_t^\theta$ to be divergence-free in order to leave $\rho_t$ invariant. Notice that this cancellation term has the form

$$[b_t^\theta]_D(x) = \sigma(x_i)\partial_t \left(\prod_{i=1}^{D-1} f_t^\theta(x_i|x_{1:i-1})\right)$$
$$= \frac{\partial}{\partial_{D-1}} \left[\sigma(x_i)\partial_t \left(F_t^\theta(x_{D-1}|x_{1:D-2}) \prod_{i=1}^{D-2} f_t^\theta(x_i|x_{1:i-1})\right)\right].$$
(22)

To ensure $b_t^\theta$ is divergence-free, we add a *compensating* term to the $D-1$-th coordinate,

$$[b_t^\theta]_{D-1}(x) = -\frac{\partial}{\partial x_D} \left[\sigma(x_i)\partial_t \left(F_t^\theta(x_{D-1}|x_{1:D-2}) \prod_{i=1}^{D-2} f_t^\theta(x_i|x_{1:i-1})\right)\right].$$
(23)

This results in a $b_t^\theta$ that is divergence-free since

$$\nabla \cdot b_t^\theta = \frac{\partial}{\partial x_D}[b_t^\theta]_D + \frac{\partial}{\partial x_{D-1}}[b_t^\theta]_{D-1} = 0.$$
(24)

However, the $b_t^\theta$ in eq. (23) introduces a new spurious flux in the $D-1$ coordinate since $[b_t^\theta]_{D-1} \neq 0$ as $x_{D-1} \to \infty$. To completely remove spurious flux while keeping $b_t^\theta$ divergence-free, we must recursively add *cancellation* and *compensating* terms to each coordinate, until every coordinate has their spurious flux removed. This results in the following vector field for the general case:

$$[b_t^\theta]_i(x) = \begin{cases} \sigma(x_i)\partial_t \left(\prod_{j=1}^{D-1} f_t^\theta(x_j)\right), & \text{if } i = D \\ -\left(\prod_{j=2}^{D} \sigma'(x_j)\right)\partial_t F_t^\theta(x_1), & \text{if } i = 1 \\ \left(\prod_{j=i+1}^{D} \sigma'(x_j)\right)\left(\sigma(x_i) - f_t^\theta(x_i|x_{1:i-1})\right)\partial_t \left(\prod_{j=1}^{i-1} f_t^\theta(x_j|x_{1:j-1})\right) & \\ \quad - \left(\prod_{j=i+1}^{D} \sigma'(x_j)\right)\partial_t f_t^\theta(x_i|x_{1:i-1})\left(\prod_{j=1}^{i-1} f_t^\theta(x_j|x_{1:j-1})\right), & \text{otherwise} \end{cases}$$
(25)

The following results show that the $b_t^\theta$ in eq. (25) is divergence-free and that it completely removes the spurious flux problem.

**Lemma 3.** *The vector field $b_t^\theta$ in eq. (25) is divergence-free, i.e. $\nabla \cdot b_t^\theta = 0$.*

**Theorem 1.** *Let $\rho_t$ and $j_t$ be given by eq. (6) and eq. (7), respectively, with $a_\theta^t$ given by eq. (12) and $b_t^\theta$ by eq. (25). Then the continuity eq. (5) holds, the density satisfies $\rho_t > 0$ and $\int_{\mathbb{R}^D} \rho_t(x) dx = 1$, and in addition there are no spurious flux, i.e. $j_t \to 0$ as $x \to \infty$.*

Note that in our implementation, we compute all quantities in eq. (25) in parallel across all coordinates using autoregressive architectures, and in logarithm space for numerical stability. The derivatives $\partial_t$ are computed using memory-efficient forward-mode automatic differentiation, so the total cost of computing eq. (25) is on par with a single forward evaluation of the autoregressive model.

### 3.6 THE FACTORIZED CASE: SIMPLIFICATIONS AND GENERALIZATIONS

The vector field in eq. (25) can be drastically simplified for the factorized case by setting $\sigma(x_i) = F_t^\theta(x_i)$, which gives

$$
[b_t^\theta]_i(x) = \begin{cases} F_t^\theta(x_D) \partial_t \left( \prod_{j=1}^{D-1} f_t^\theta(x_j) \right), & \text{if } i = D \\ -\left( \prod_{j=2}^{D} f_t^\theta(x_j) \right) \partial_t F_t^\theta(x_1), & \text{if } i = 1 \\ -\left( \prod_{j=i+1}^{D} f_t^\theta(x_j) \right) \partial_t F_t^\theta(x_i) \left( \prod_{j=1}^{i-1} f_t^\theta(x_j) \right), & \text{otherwise.} \end{cases} \tag{26}
$$

Substituting this back into eq. (9) results in the simplified velocity field (for $g_t = 0$):

$$
[u_t^\theta]_i(x) = j_t^\theta(x)/\rho_t^\theta(x) = (-\partial_t a_t^\theta(x) + b_t^\theta(x))/\rho_t^\theta(x) = -\frac{\partial_t F_t^\theta(x_i)}{f_t^\theta(x_i)} \tag{27}
$$

for all $i \in \{1, \dots, D\}$, which is easy to implement and compute in practice. Furthermore, we note that for the factorized model, the velocity is always *kinetic optimal* as $u_t(x)$ in eq. (27) is a gradient field. In particular, it means that it is the velocity that results in the shortest paths out of all velocities that generate this $\rho_t$.

To increase the flexibility of the factorized model, note that we can combine multiple pairs of $(\rho_t^k, u_t^k)$ into a mixture model with coefficients $\gamma^k$:

$$
\rho_t(x) = \sum_{k=1}^{K} \gamma^k \rho_t^k(x), \qquad u_t(x) = \sum_{k=1}^{K} \frac{\gamma^k \rho_t^k(x)}{\rho_t(x)} u_t^k(x). \tag{28}
$$

**Proposition 1.** *If each pair of $\rho_t^k$ and $u_t^k$ satisfy the Fokker-Planck equation as in eq. (2), then the $\rho_t$ and $u_t$ as defined in eq. (28) also satisfy the Fokker-Planck equation. Proof is provided in Appendix B.*

### 3.7 LEARNING AN INDEPENDENT DIVERGENCE-FREE COMPONENT

While the choice of $b_t^\theta$ in eq. (25) removes the spurious nonzero flux values at infinity, this parameterization lacks the flexibility in optimizing $j_t$, *e.g.* it does not necessarily correspond to kinetic optimal velocity fields $u_t$, unless using the factorized model discussed in Section 3.6. In order to handle a wider range of applications where we do optimize over $u_t$, we can include a flexible learnable component into $j_t$ that leaves the continuity equation invariant.

Let the new flux field be parameterized as

$$
j_t = -\partial_t a_t^\theta + b_t^\theta + v_t^\theta, \qquad \text{where } \nabla \cdot v_t^\theta = 0, \tag{29}
$$

so $f_t^\theta : \mathbb{R}^{D+1} \to \mathbb{R}^D$ is a divergence-free vector field. This construction still satisfies eq. (5) because

$$
\partial_t \rho_t + \nabla \cdot j_t = \partial_t (\nabla \cdot a_t^\theta) - \nabla \cdot (\partial_t a_t^\theta + b_t^\theta + v_t^\theta) = 0 \tag{30}
$$

To satisfy the divergence-free constraint, we adopt the construction in Richter-Powell et al. (2022) and parameterize an matrix-valued function $A_t^\theta : \mathbb{R}^{D+1} \to \mathbb{R}^{D \times D}$ with neural networks and we let

$$
v_t^\theta = \nabla \cdot \left( A_t^\theta - (A_t^\theta)^T \right) \tag{31}
$$

| Model | Pinwheel | Earthquakes JP | COVID-19 NJ | CitiBike |
|---|---|---|---|---|
| Conditional KDE (Chen et al. (2020)) | 2.958 $_{\pm 0.000}$ | 2.259 $_{\pm 0.001}$ | 2.583 $_{\pm 0.000}$ | 2.856 $_{\pm 0.000}$ |
| Neural Flow (Biloš et al. (2021)) | N/A | 1.633 | 1.916 | 2.280 |
| CNF (Chen et al. (2020)) | 2.185 $_{\pm 0.003}$ | 1.459 $_{\pm 0.016}$ | 2.002 $_{\pm 0.002}$ | 2.132 $_{\pm 0.012}$ |
| NCL++ (Factorized) | 2.028 $_{\pm 0.062}$ | 1.217 $_{\pm 0.024}$ | 1.846 $_{\pm 0.012}$ | 1.462 $_{\pm 0.033}$ |
| NCL++ (Autoregressive) | **1.936** $_{\pm 0.083}$ | **1.184** $_{\pm 0.031}$ | **1.732** $_{\pm 0.009}$ | **1.239** $_{\pm 0.024}$ |

Table 1: Negative log-likelihood per event on held-out test data (lower is better).

where the divergence is taken over the rows of the anti-symmetric matrix $A_t^\theta - (A_t^\theta)^T$. Let $A_{t;i,j}^\theta$ denote the $(i, j)$ entry of $A_t^\theta$. We can easily verify that $v_t^\theta$ is divergence-free with the following:

$$\nabla \cdot v_t^\theta = \sum_{i=1}^{D} \sum_{j=1}^{D} \partial_{x_i} \partial_{x_j} \left( A_{t;i,j}^\theta - (A_{t;i,j}^\theta)^T \right) - \partial_{x_j} \partial_{x_i} \left( A_{t;i,j}^\theta - (A_{t;i,j}^\theta)^T \right) = 0 \qquad (32)$$

## 4 EXPERIMENTS

In each of the following sections, we consider broader and broader problem statements, where each successive problem setting roughly builds on top of the previous ones. Throughout, we parameterize the density $\rho_t$ and a flux $j_t$ following Section 3.1 in order to satisfy the continuity equation, and compute the velocity field $u_t$ using eq. (9). All models are trained without simulating the differential equation in eq. (4). While there exist simulation-free baselines for the first few settings (Sections 4.1 & 4.2), to the best of our knowledge, we are the first truly simulation-free approach for the more complex problem setting involving mean-field optimal control (Section 4.3). Experimental details are provided in Appendix C.

### 4.1 SPATIO-TEMPORAL GENERATIVE MODELING

Our goal is to fit the model to data observations from an unknown data distribution $q(t, x)$. We consider the unconditional case of generative modeling where samples are obtained from marginal distributions across time, while the individual trajectories are unavailable. As a canonical choice, we use the cross entropy as the loss function for learning $\rho_t$.

$$L_{\text{GM}} = \mathbb{E}_{t, x \sim q(t, x)} \left[ -\log \rho_t(x) \right] \qquad (33)$$

We consider datasets of spatial-temporal events preprocessed by Chen et al. (2020) and these datasets are sampled randomly in continuous time. We take only the spatial component of these datasets, as this is our core contribution. To evaluate the capability of our method on modeling spatial-temporal processes, we test our proposed method on these datasets and compare against state-of-the-art models on these datasets by Chen et al. (2020) and Biloš et al. (2021).

We report the log-likelihoods per event on held-out test data of our method and baseline methods in Table 1, highlighting that our method outperforms the baselines with substantially better spatial log-likelihoods across all datasets considered here.

### 4.2 LEARNING TO TRANSPORT WITH OPTIMALITY CONDITIONS

We next consider settings where the data are only sparse observed at select time values, and the goal is to learn a transport between each consecutive observed time values, subject to some optimality conditions. The simplest case is dynamic optimal transport Villani (2021), where we introduce a kinetic energy to the loss function in order to recover short trajectories between consecutive time values.

$$L_{\text{OT}} = \sum_{t \in \{t_i\}_{i=1}^{n}} \mathbb{E}_{x \sim q_{t_i}(x)} \left[ -\log \rho_t(x) \right] + \int_{t_0}^{t_n} \mathbb{E}_{x \sim \rho_t(x)} \left[ \|u_t(x)\|^2 \right] dt \qquad (34)$$

As our benchmark problem, we investigate the dynamics of cells based on limited observations, focusing on the single-cell RNA sequencing data of embryoid bodies as analyzed by Neklyudov et al.

| Model | $W_2(q_{t_1}, \widehat{q}_{t_1})$ | $W_2(q_{t_2}, \widehat{q}_{t_2})$ | $W_2(q_{t_3}, \widehat{q}_{t_3})$ | $W_2(q_{t_4}, \widehat{q}_{t_4})$ |
|---|---|---|---|---|
| OT-flow | 0.75 | 0.93 | 0.93 | 0.88 |
| Entropic Action Matching | 0.58 $_{\pm 0.015}$ | 0.77 $_{\pm 0.016}$ | **0.72** $_{\pm 0.007}$ | 0.74$_{\pm 0.017}$ |
| Neural SDE | 0.62 $_{\pm 0.016}$ | 0.78 $_{\pm 0.021}$ | 0.77 $_{\pm 0.017}$ | 0.75 $_{\pm 0.017}$ |
| NCL++ (Factorized, Directly Sampled) | 0.56 $_{\pm 0.009}$ | 0.79 $_{\pm 0.012}$ | 0.74 $_{\pm 0.010}$ | 0.72 $_{\pm 0.006}$ |
| NCL++ (Autoregressive, Directly Sampled) | **0.52** $_{\pm 0.004}$ | **0.74** $_{\pm 0.005}$ | **0.72** $_{\pm 0.003}$ | **0.69** $_{\pm 0.004}$ |
| NCL++ (Factorized, Transported) | 0.58 $_{\pm 0.015}$ | 0.80 $_{\pm 0.007}$ | 0.76 $_{\pm 0.009}$ | 0.75 $_{\pm 0.009}$ |
| NCL++ (Autoregressive, Transported) | 0.53 $_{\pm 0.013}$ | 0.76 $_{\pm 0.008}$ | 0.73 $_{\pm 0.005}$ | 0.71 $_{\pm 0.008}$ |

Table 2: The Wasserstein-2 distance between the test marginals and marginal distributions from the model calculated by the test samples and the samples obtained from directly sampling from the model. For our own methods, we report standard deviation estimated across 20 runs.

| Model | $W_2(q_{t_1}, \widehat{q}_{t_2})$ | $W_2(q_{t_2}, \widehat{q}_{t_3})$ | $W_2(q_{t_3}, \widehat{q}_{t_4})$ |
|---|---|---|---|
| NCL++ (Factorized) | 3.45 $_{\pm 0.125}$ | 3.67 $_{\pm 0.103}$ | 4.09 $_{\pm 0.147}$ |
| NCL++ (Autoregressive) | **2.85** $_{\pm 0.075}$ | **3.14** $_{\pm 0.082}$ | **3.62** $_{\pm 0.097}$ |

Table 3: The Wasserstein-2 distance between the distributions transported from the test marginals at $t_i$ and the test marginals at $t_{i+1}$. We use this Wasserstein-2 distance to measure how kinetically optimal our trained maps are. We report the mean and standard deviation estimated across 20 runs.

(2023). This dataset offers sparse observations in a 5-dimensional PCA decomposition of the original cell data introduced by Moon et al. (2019) at discrete time points $t_0 = 0, t_1 = 1, t_2 = 2, t_3 = 3, t_4 = 4$. Our objective is twofold: first, to fit the time-continuous distribution of the dataset with given sparse observations, and second, to obtain optimal transport (OT) paths between these marginal distributions.

To evaluate the performance of our model on this problem setup, we compute the Wasserstein-2 distance between our fitted model $\rho_t$ and the data distribution $q_t$ at $t = 0, 1, 2, 3, 4$. The Wasserstein-2 distance is computed with the samples we directly sample from the model marginal distribution $\rho_t$ and the held-out test data from the dataset. Additionally, we compute the Wasserstein-2 distance between test marginals and model marginals by transporting samples from the data marginal $q_{t_i}$ to our estimated marginals at the next time value $\widehat{q}_{t_{i+1}}$ using the trained velocity field.

Compared to prior works, we not only learn the transport map and the marginal densities of the dataset, but also optimize the model for the kinetically optimal transport map. Our model has the flexibility in terms of training for the kinetic optimal transport map because of the additional learnable component $v_t^\theta$ that can be incorporated into the flux term (Section 3.7), all the while capable to be learned without sequential simulation of the underlying dynamical system. Results of optimizing for kinetically optimal transport map are reported in Table 3. As compared to the factorized model, which is simpler and easier to train, the autoregressive model achieves better performance in both density fitting and optimizing for the optimal transport.

## 4.3 MEAN-FIELD STOCHASTIC OPTIMAL CONTROL

Stochastic optimal control (SOC; Mortensen 1989; Fleming & Rishel 2012; Kappen 2005) aims at finding the optimal dynamics model given an objective function, instead of data observations. SOC problems arise in wide variety of applications in sciences and engineering (Pham, 2009; Fleming & Stein, 2004; Zhang & Chen, 2022; Holdijk et al., 2023; Hartmann et al., 2013; 2017) and we provide numerical evidence to illustrate that our framework can be extended to solving SOC problems, including mean-field type of SOC problems (Bensoussan et al., 2013), which have wide applications in finance (Fleming & Stein, 2004; Pham, 2009; Aghion, 1990) and robotics (Theodorou et al., 2011; Pavlov et al., 2018). Reducing the SOC problem into our setting in eq. (1), we have the following

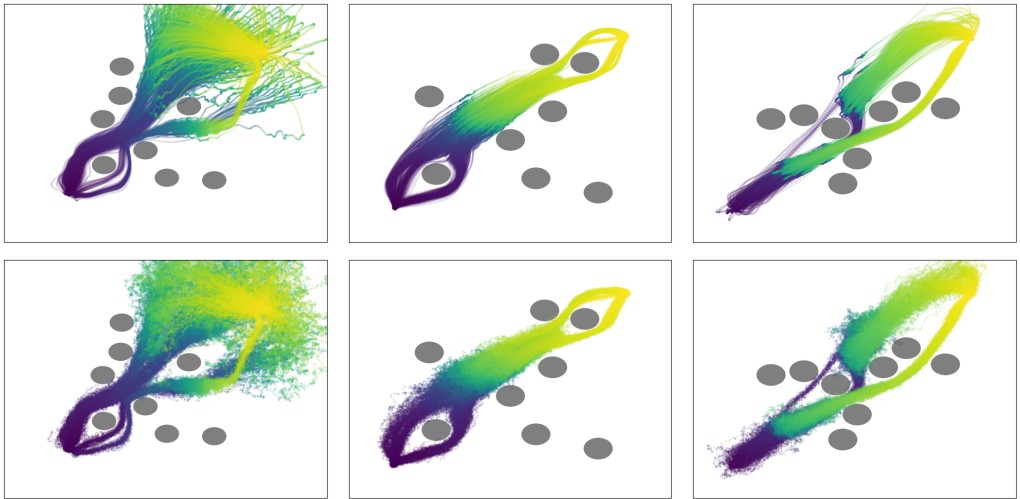

Figure 2: Transport paths of a trained factorized model on the motion planning task of different environments with randomly initialized circular obstacles. We train the model with a diffusion coefficient $g_t = 0.0$ and we sample the model via solving eq. (4) with $g_t = 0.0$ (first row) and $g_t = 0.5$ (second row). Note that in the case of $g_t = 0.0$, eq. (4) reduces to a deterministic ODE.

objective function:

$$L_{\text{SOC}} = \int_0^1 \phi_t(\rho_t)dt + \mathbb{E}_{x \sim \rho_t} \left[ \frac{1}{2\sigma_t^2} \|u_t(x) - v_t(x)\|^2 \right] dt + \Phi(\rho_1) + \mathbb{E}_{x_0 \sim q_0} \left[ -\log \rho_0(x_0) \right] \tag{35}$$

where $q_0$ is a given initial distribution, $v_t$ is a given base drift function, and we use $\Phi(\rho_1) = \mathbb{E}_{x_1 \sim q_1} \left[ -\log \rho_1(x_1) \right]$ as the terminal cost so that the model can also be fitted to a given terminal distribution $q_1$. For our task, we formulate problems with circular obstacles that the model must navigate around. In particular, for circular obstacles with radius $R$ and center coordinate $c$, the running cost is defined as:

$$\int_0^1 \phi_t(\rho_t)dt = \mathbb{E}_{X_t \sim \rho_t}[\text{softplus}\left( R^2 - (X_t - c)^2 \right)] + \eta \mathbb{E}_{X_t \sim \rho_t}[\log \rho_t(X_t)] \tag{36}$$

where $\mathbb{E}_{X_t \sim \rho_t}[\log \rho_t(X_t)]$ is the entropy of the model—*i.e.*, a mean-field cost—used to encourage the model to find all the possible paths and $\eta$ is a weighting.

We test our method on the motion planning tasks introduced by Le et al. (2023). The task is to navigate from the source to the target distribution while avoiding randomly initialized circular obstacles, where we use the entropy regularization to encourage finding multiple paths and to ensure we find robust solutions. We visualize the trained model in Figure 2, where our framework trained with diffusion coefficient $g_t = 0$ can handle different environments and can also be used to produce reasonable samples when additional noise is present, *i.e.*, $g_t > 0$.

## 5 CONCLUSION

We propose a simulation-free framework for training continuous-time stochastic processes over a large range of objectives, by combining Neural Conservation Laws with likelihood-based models. We demonstrated the flexibility and capacities of our method on various applications, including spatio-temporal generative modeling, learning optimal transport between arbitrary densities, and mean-field stochastic optimal control. Especially at low dimensional settings, our method easily outperforms existing methods. However, the reliance on likelihood-based models make it difficult to be scaled up to high dimensions. We acknowledge these limitations and leave them for future works.

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

## A  JACOBIAN SYMMETRIZATION LOSS

With the flexible learnable component $f_t^\theta$, we can learn the Hodge decomposition of the velocity field to remove the extra divergence-free (i.e., rotational) components of the model. As a direct consequence of the the Benamou-Brenier formula Villani (2021); Albergo & Vanden-Eijnden (2022), the velocity field that achieves the optimal transport has no divergence-free component.

A necessary and sufficient condition for a velocity field of being a gradient field is having symmetric Jacobian matrix with respect to all spatial dimensions. By the Hodge decomposition, any time-dependent velocity field $v^t : \mathbb{R}^{D+1} \to \mathbb{R}$ can be expressed as a sum of a divergence-free field and a gradient field:

$$v^t = \nabla \phi^t + \eta^t \tag{37}$$

where $\eta^t$ is divergence-free. Let $J^t : \mathbb{R}^{D+1} \to \mathbb{R}^{D \times D}$ be the Jacobian matrix of $v^t$ with respect to its spatial dimensions and we denote its $(i, j)$ entry by $J_{i,j}^t = \partial_{x_j} v_i^t$. Then, the Jacobian matrix is symmetric if and only if $\partial_{x_j} v_i^t = \partial_{x_i} v_j^t$ for any $i, j$. It follows that $\partial_{x_j} v_i^t = \partial_{x_i} v_j^t$. Consequently, $\int_{-\infty} v_i^t dx_i = \int_{-\infty} v_j^t dx_j$ for any $i, j$. Then let $\phi^t = \int_{-\infty} v_1^t dx_1$ and we obtain $v^t = \nabla \phi^t$ is a gradient field. Conversely, if $v^t = \nabla \phi^t$ is a gradient field, then $\partial_{x_j} v_i^t = \partial_i \partial_j \phi^t = \partial_{x_i} v_j^t$ and $J_{i,j}^t$ is symmetric.

Motivated by this observation of the equivalence between the symmetry of the Jacobian and the OT plan, we train $f_t^\theta$ with the loss

$$L_{\text{OT}} = \mathbb{E}_{t, x \sim \rho_t(x)} ||J_t^\theta(x) - (J_t^\theta)^T(x)||_F \tag{38}$$

where $|| \cdot ||_F$ denotes the Frobenius norm. We can compute this loss by using the Hutchinson trace estimator Hutchinson (1989). Let $v \sim \mathcal{N}(0, I)$ be a random Gaussian vector and $u_t(x) = v^T J_t^\theta(x)$ and $w_t(x) = J_t^\theta(x)v$. Computing $u$ and $w$ requires one vector-Jacobian product (VJP) and one Jacobian-vector product (JVP), respectively. Therefore,

$$
\begin{aligned}
L_{\text{OT}} &= \mathbb{E}_{t, x \sim \rho_t(x)} ||J_t^\theta(x) - (J_t^\theta)^T(x)||_F \\
&= \mathbb{E}_{v \sim \mathcal{N}(0, I); t, x \sim \rho_t(x)} [v^T \left( J_t^\theta(x) - (J_t^\theta)^T(x) \right) \left( J_t^\theta(x) - (J_t^\theta)^T(x) \right) v] \\
&= \mathbb{E}_{v \sim \mathcal{N}(0, I); t, x \sim \rho_t(x)} [u_t^T(x) w_t(x) - 2 w_t^T(x) w_t(x) + w_t^T(x) u_t(x)]
\end{aligned}
\tag{39}
$$

So, the stochastic estimation of the loss takes only one VJP and one JVP at each sample, which is computationally feasible even in high dimensions.

While this could arguably be better than regularizing kinetic energy for finding kinetic optimal solutions, as it no longer requires an explicit tradeoff between kinetic energy and the other cost functions, we did not observe a meaningful improvement over simply regularizing kinetic energy.

## B  PROOF OF PROPOSITION 1

*Proof.* We check that $\rho_t$ and $u_t$ satisfy eq. (2):

$$
\begin{aligned}
\partial_t \rho_t &= \sum_{k=1}^K \gamma^k \left( \partial_t \rho_t^k \right) = \sum_{k=1}^K \gamma^k \left( -\nabla \cdot (u_t^k \rho_t^k) + \tfrac{1}{2} g_t^2 \Delta \rho_t^k \right) \\
&= -\nabla \cdot \left( \sum_{k=1}^K \frac{\gamma^k \rho_t^k}{\rho_t} u_t^k \right) \rho_t + \tfrac{1}{2} g_t^2 \Delta \sum_{k=1}^K \gamma^k \rho_t^k = -\nabla \cdot u_t \rho_t + \tfrac{1}{2} g_t^2 \Delta \rho_t
\end{aligned}
\tag{40}
$$

$\square$

## C  EXPERIMENTAL SETUP

**Neural Network Architecture**  For training the autoregressive model, we use the MADE architecture (Germain et al., 2015) with sinusoidal time embeddings of width 128 (Tancik et al., 2020). For the neural networks we use to parameterize the mean and the scale of both the autoregressive model and the factorized model, we pass the input first into the sinusoidal time embeddings before feeding into a four-layer MLP of hidden dimension 256 on each layer.

**Training Details**     For all the numerical experiments we present, we use a learning rate of $3e - 4$ with the Adam optimizer (Kingma, 2014) and a cosine annealing learning rate scheduler.

**Spatio-temporal Generative Modeling**     The total number of iterations we run for the experiments are generally $10^3$ epochs with a batch size of 256. We found that the training is stable with a simple four-layer MLP parametrization for the mean and the scale of the mixtures of factorized logistics. Also, the MLP parameterization along with the mixture combinations in the factorzied model turned out to be expressive enough for the experiments we have explored.

**Learning To Transport With Optimality Conditions**     For the single-cell RNA sequence dataset used in (Moon et al., 2019), we find that both the factorized model and the autoregressive model will easily get overfitted if we use more than 64 modes in the mixture. For the numerical results we are reporting, we use mixtures of size $L = 16$ for each of the coordinates in the autoregressive model (14), and we use a mixture of size $K = 32$ for the factorized model (28). Also, we find that having the term $\int_{t_0}^{t_n} \mathbb{E}_{x \sim \rho_t(x)} \left[ \|u_t(x)\|^2 \right] dt$ in the loss objective is extremely helpful for both finding the kinetic optimal paths and preventing overfitting.

**Mean-field Stochastic Optimal Control**     To achieve consistent results for this experiment, we train the objective function $L_{\text{SOC}}$ by gradually introducing different terms in it. We first train the log-likelihood term $\mathbb{E}_{x_0 \sim q_0}[-\log \rho_0(x_0)] + \mathbb{E}_{x_1 \sim q_1}[-\log \rho_1(x_1)]$ for $10^3$ iterations with a batch size of 512. Then, we introduce the term $\mathbb{E}_{x \sim \rho_t} \left[ \frac{1}{2\sigma_t^2} \|u_t(x) - v_t(x)\|^2 \right] dt$ for another $10^3$ iterations. Finally, we introduce the running cost $\int_0^1 \phi_t(\rho_t) dt$ and train for $2 \times 10^4$ iterations. This training technique helps stablizes the training.

# D   COMPARISON TO SIMULATION-BASED METHODS

We conduct the experiments outlined in Section 4.1 using the autoregressive model from our method and time-varying continuous normalizing flows (CNFs) Chen et al. (2019b), with the implementation from Chen et al. (2020). The wall-clock training time is 18 minutes for our method, compared to 133 minutes for the time-varying CNFs. Notably, our approach outperforms the time-varying CNFs despite having a simpler architecture and fewer parameters in the neural network. Additionally, our method is simulation-free, avoiding the need to solve an ODE or adjoint equations during training, which can be costly both in terms of simulation time and backpropagation through the simulation trajectories.

Furthermore, we observe that our method requires significantly less memory than the time-varying CNFs, as it does not need to store simulation trajectories or maintain computation graphs along these trajectories. This memory efficiency enables us to use much larger batch sizes during training.

