# OpenReview forum: "Simulation-Free Differential Dynamics through Neural Conservation Laws"
_ICLR.cc/2025/Conference — Submitted to ICLR 2025_

### Official Review · Reviewer_k3XS · 2024-10-31

**Soundness:** 3
**Presentation:** 3
**Contribution:** 3
**Rating:** 6
**Confidence:** 3

**Summary:**

The paper presents a novel framework for the simulation-free training continuous-time diffusion processes in a general setting. This is achieved by a coupled parameterization that jointly models the *density* path $\rho_t$  and associated *dynamics* $u_t$ (vector field). The authors achieve this by enforcing the Fokker-Plank (FP) equation as a hard constraint for a very general objective function.  The  FP equation ensures that $dX_t =u_t(x_t)dt + g_tdW_t$ has marginally density $X_t\sim p_t$. Enforcing these constraints hardly allows the application standard unconstrained optimization. The authors test the proposed method on a set of tasks, including: density estimation, optimal transport, and mean-field optimal control.

**Strengths:**

1. **Novelty**: The paper presents an interesting method that, to my knowledge, is novel.
2. **Clarity**: The paper is generally well-organized, with a logical progression from theoretical developments to practical implementation. It elaborates on design choices made by the authors, its implications and how to adress potential problems (i.e., spurious flux).
3. **Generality**: The paper demonstrates that the method can be applied to various problem settings. The authors show that the method can achieve state-of-the-art results on some (lower-dimensional!) spatio-temporal generative modeling data sets and on learning transport maps in cellular dynamics.

**Weaknesses:**

1. **Scalability**: While promising in low-dimensional settings, the authors acknowledge limitations in scaling this methodology to higher dimensions "due to its reliance on likelihood-based models" and leave it to future work. Yet many other likelihood-based generative models (discrete normalizing flows, autoregressive models (transformer, RNN), etc.) are scalable to (moderately) high dimensions. I think that the paper would benefit from at least a demonstration that the proposed method can scale (e.g. standard MNIST, CIFAR generative modeling tasks).

**Questions:**

Can the authors provide a demonstration of scalability (see Weakness)?

---

> ### Author Response · Authors · 2024-11-27
>
> We thank the reviewers for the helpful comments. Here's our reply to the concern related to the scalability of our method:
>
> The reviewer is correct that our method is only as scalable as autoregressive (AR) models, due to its reliance on AR models for scaling to high-dimensional data. However, note that AR models are already used at scale for text, e.g. chatGPT, with significant engineering efforts such as KV caching significantly speeding up AR models. We unfortunately do not have the same expertise to enable fast AR models and used only small MLPs for our experiments. As such, we focused mainly on low-dimensional problem settings. Despite this, we have shown that there is significant promise in using our approach, outperforming all existing methods on the data sets that we considered. This signifies that it is worth spending significant effort in scaling up this approach with a fast AR model implementation for continuous space, which is a research question of its own. We note, however, that our proposed contributions are orthogonal to the choice of AR model.

---

> > ### Comment · Reviewer_k3XS · 2024-11-28
> >
> > I thank the authors for their response. I will maintain my previous assessment.

---

### Official Review · Reviewer_71qs · 2024-11-02

**Soundness:** 3
**Presentation:** 3
**Contribution:** 3
**Rating:** 6
**Confidence:** 2

**Summary:**

The manuscript introduces a simulation free approach for the training of diffusion processes. This is achieved by an ansatz the exactly satisfies the Fokker-Planck equation -- which circumvents costly simulation for its solution. The approach is motivated by Neural Conservation Laws. The authors then numerically demonstrate their method.

Disclaimer: The paper is somewhat outside of my comfort zone, so my judgment is to be taken with a grain of salt.

**Strengths:**

- Hard-coding PDEs or PDE terms often outperforms soft penalties (as would be prevalent in physics-informed neural networks). Constructing an ansatz that does this for a Fokker-Planck equation is thus quite interesting.

- The paper is well-written and more or less understandable, even for me as a reader outside of the field.

**Weaknesses:**

- I am unfortunately not very familiar with the challenges or difficulties of the field, so I will pose questions rather than pointing out weaknesses.

**Questions:**

- Universality of the ansatz: In equations (1)-(3) the authors are interested in finding the optimal $u$ and $\rho$ subject to the Fokker-Planck equation. To avoid the solution of the Fokker-Planck equation, both $u$ and $\rho$ are parametrized in terms of a vector field $a_\theta$ (and also $b_\theta$). Can the authors comment on the universality of that construction? Will every pair $(u, \rho)$ be asymptotically realizable? If no, which pairs can be reached and which cannot? This can be a potential problem for the proposed method if the minimizers of problem (1)-(3) are not well representable.

Minor Questions/Typos:

- Line 131 The flux $j_t$ should be time dependent, hence map $\mathbb R^{1+D} \to \mathbb R^D$, correct?

---

> ### Author Response · Authors · 2024-11-27
>
> We sincerely thank the reviewer for their constructive feedback. We will address reviewer's questions below.
>
> **Question 1 (universality)**: Note that since mixture models are universal density approximator, both the autoregressive and mixture of factorized ansatzes can universally approximate any $\rho_t$. The question is then whether the framework can approximate any velocity $u_t$. Note that by the Fokker-Planck equation, the velocity field is only determined by $\rho_t$ up to a divergence-free component. Since we also model a divergence-free component in our model (Section 3.7), our model can universally approximate any pair of reasonable $(u_t, \rho_t)$ as long as our divergence-free model is universal, which is a result proven in [1].
>
> [1] “Neural Conservation Laws: A Divergence-Free Perspective” Richter-Powell et al. (2022)
>
> **Question 2 (typo)**: We thank the reviewer for helping us find this typo. We corrected it in the revised draft.

---

> > ### Comment · Reviewer_71qs · 2024-12-03
> >
> > Thank you for your reply. I keep my score and stress that my assessment should not strongly influence the acceptance decision.

---

### Official Review · Reviewer_hL5b · 2024-11-07

**Soundness:** 2
**Presentation:** 2
**Contribution:** 3
**Rating:** 5
**Confidence:** 2

**Summary:**

The paper introduces a method to train a diffusion process that satisfies the Fokker-Planck equation along with a "sum-to-1" density constraint, transforming this constrained optimization problem into an unconstrained one through reparameterization. This approach can cause the spurious flux phenomenon, which can be mitigated by a careful choice of parameters in the reparameterized process.

**Strengths:**

* The idea of converting the constrained problem into an unconstrained one is nice.
* The method is also theoretically justified, although I have not fully checked the math.
* The proposed approach is "simulation-free", which I think should translate to better cost-accuracy tradeoff.

**Weaknesses:**

### Major
* The method claims scalability (i.e. better cost-accuracy trade-off) as a primary benefit, yet there is no such comparison (experimental or otherwise). A comparison of the wall-clock time or FLOPS will be good.
* While this is the first simulation-free approach to address Section 4.3, is there a non-simulation-free baseline to check the quality of the solution?
* Justifications for Lemma 3 and Theorem 1 are missing. Remarks on Proposition 1 would be helpful, although this might be trivial for experts in this area.
### Minor
* Missing standard errors for baselines in Tables 1 & 2.
* Detailed hyperparameters would support reproducibility.
* Minor typos and grammar errors (e.g., line 228 "functions dependent on $t$").

**Questions:**

* How is "negative log-likelihood" defined in Table 1? Is it (33)?
* Are there any recent baselines for the setup in Section 4.1?

---

> ### Author Response · Authors · 2024-11-27
>
> We thank the reviewer for the careful review and here are the replies to the concerns of the reviewer. Here are our replies:
>
> **Weakness 1& 2 (cost-accuracy trade-off and comparison to simulation-based methods)**:We provide in Appendix D a new discussion around the wallclock time requirement for our method and the simulation-based approach for training CNFs. Of note, we only use 13% of the wallclock time of the simulation-based method.
>
> **Weakness 3 (justification of theorems and lemmas)**: We would like to provide more clarifications on these theorems and lemmas, but we note that these are direct consequences from the construction of the density $\rho_t^\theta$ and the spurious flux cancellation terms $b_t^\theta$. We will be happy to clarify more or provide detailed derivations in the future replies.
>
> **Weakness 4 (missing standard errors for baselines)**: We report the baselines directly from the cited papers, and the missing standard errors for these rows were not provided in the original sources.
>
> **Weakness 5 (typos)**: Thanks for pointing out these typos. We have corrected them in the revised draft.
>
> **Weakness 6 (implementation details)**: We have provided some details in Appendix C and we will release reproducible code upon paper publication.
>
> **Question 1 (negative log-likelihood)**: Yes, it is defined as in Equation (33).
>
> **Question 2 (recent baselines)**: We would like to clarify that, to the best of our knowledge, the listed baselines are the most recent ones for the datasets we consider. Additionally, in Section 4.2, we compare our framework with Action Matching, which was published in 2023.
>
> Given these additional explanations about our work and the experimental evaluations we have added, we respectfully ask the reviewer to consider raising the score to better reflect our work's contributions.

---

> > ### Comment · Reviewer_hL5b · 2024-11-29
> >
> > Thank you for your response. I am not an expert in this field, so it is best for the other reviewers to judge the novelty of the approach. From an outsider's perspective, I do think the proposed approach is quite interesting.
> >
> > My main concerns are with the presentation of the results and the execution of the experiments. For example:
> > 1. Why is there only a wall-clock time comparison for Section 4.1, but not for Sections 4.2 and 4.3? Why not present this in a tabular format, which is much easier to read and interpret?
> > 2. I also believe it is best practice to reproduce the baselines on your own hardware rather than copy-pasting from previous work. Also, how would you interpret the standard error if the number of repetitions differs across methods?
> >
> > I believe this paper is worth attention and have raised my score to 5. This work will definitely get a higher score if the experiments are further improved.

---

### Official Review · Reviewer_kXdF · 2024-11-08

**Soundness:** 2
**Presentation:** 3
**Contribution:** 2
**Rating:** 5
**Confidence:** 4

**Summary:**

The paper proposes a new simulation-free method for training (stochastic) dynamics which is driven by a continuity equation (i.e., follows stochastic sde) and satisfies some objectives.
In particular, the authors propose specific parameterization of the class of continuity equations and then optimize the given objectives in this class.

**Strengths:**

Personally, I like the text. It is more-or-less clearly written, understandable (but somewhere in pages 5-6 some inconsistencies appear with indices ($i \leftrightarrow D$) - I reflected them in my questions / comments); logic is clear. So, I thank the authors (but some of the “raw” moments need tweaking).

Also, It was a pleasant nostalgia for me, when reading about all of these autoregressive models and mixtures of logistics.

**Weaknesses:**

1. The main question for the method (and it is partially reflected in the appendix) is that the approach is not dimension-scalable. Also, the proposed parameterization with autoregressive models and mixture of logistics does not seem to be very expressive.
2. No source code for the submission.
3. While text is clearly written, some "raw" moments and misprints to be fixed (se below).

Also, please see my questions and comments below

**Questions:**

1. Lines 037 - 038. “[...]but these methods all require simulating from the learned diffusion process to some varying degrees [Peyre&Cuturi]” - what do you mean by citing this work here?
2. A general question, eq. 12. The choice for $a_t^{\theta}$ looks a bit strange - it is kind of “not symmetric”, not “invariant” under the coordinates of the vector - because it “respects” only the last coordinate. Why do you consider exactly this “not-symmetric” parameterization? (I understand that formally this is correct)
3. I do not completely understand the second row of Figure 1. As I understand, black arrows on the canvases denote fluxes $j_t$. But following the definition of $a_t^{\theta}$, eq. 12 all these arrows should be directed along one axis (i.e., they should be parallel to unit vector $(0, \dots, 0, 1)$). But it seems that the actual arrows in Figure 1 may be directed along *two* axis.
4. Lines 297-298. Something strange, may be an inaccurate denotations' intersection: you add a quantity with $\Pi_{i = 1}^{D - 1}$ to the $i$-th coordinate. The same index $i$. Given my comment 2, may be $i$ should be equal to $D$? The same problem seems to affect eqs. 21, 22, 23 (it seems that somewhere $x_i$ should be substituted with $x_D$).
5. Lines 299  - 300. Do not understand the comment that $\sigma$ could be substituted with “function/NN whose first derivative has compact support. I think - for such an NN  there should be a condition, that at $x_D \rightarrow \infty$ this NN should approach $1$.
6. Results in Table 1. For CNF, you grab the results only for “time-varying CNF”. The other variants of CNF, e.g., jump CNF and attentive CNF are excluded from the comparison (and, by the way, achieve better NLL). Why?
7. Lines 468 - 475. I do not understand, how you achieve kinetic optimality (for AR model). Ok, you have an additional term introduced in Section 3.7 - but it is just improves expressivity and does not lead to kinetic optimality out of the box (as I understand)
8. Results in Figure 2. What is the reason of the “explosion” of particles at the final time points in the first column of the Figure?

**Misprints**

1. Eqs. 14, 15: For cdf $F_t^{\theta}$ and pdf $f_t^{\theta}$ - missed conditioning $\vert x_{1:i-1}$. So does for eqs. (19) and (20), (21), (22), (23), (25).
2. Eq. (22) - $\frac{\partial}{d \partial}$.

**Comments**

1. You state that your work stems from the ideas of NCL in [Richter-Powell]. However, in the manuscript, there are no comparison between the constructions proposed in your work and in [Richter-Powell]. It would be great to have such a comparison, especially how do all these diff forms in [Richter-Powell] relate to your much simpler construction.
2. I think, lemma 2 holds only for $x_i = x_D$ ($i = D$), and other coordinates of $x$ being within support of $\rho_t$ (otherwise, right-hand side of eq. (20) would be zero).
3. Lemma 3, while explained intuitively, should have rigorous proof (e.g., in appendix)
4. Section 3.7. It is bad to call your newly introduced learned divergence-free vector field to be $f_t^{\theta}$. Because $f_t^{\theta}$ is already reserved for density in the previous text.

---

> ### Author Response · Authors · 2024-11-27
>
> We thank the reviewer for the careful review and invaluable feedback to our paper. These constructive comments helped us to improve our paper. Here are our replies to the concerns raised by the reviewer.
>
> **Weakness 1 (Scalability)**: The reviewer is correct that our method is only as scalable as autoregressive (AR) models, due to its reliance on AR models for scaling to high-dimensional data. However, note that AR models are already used at scale for text, e.g. chatGPT, with significant engineering efforts such as KV caching significantly speeding up AR models. We unfortunately do not have the same expertise to enable fast AR models and used only small MLPs for our experiments. As such, we focused mainly on low-dimensional problem settings. Despite this, we have shown that there is significant promise in using our approach, outperforming all existing methods on the data sets that we considered. This signifies that it is worth spending significant effort in scaling up this approach with a fast AR model implementation for continuous space, which is a research question of its own. We note, however, that our proposed contributions are orthogonal to the choice of AR model.
>
> **Weakness 2 (Source Code)**: Yes, we plan on open sourcing after publication.
>
> **Weakness 3 (Misprints and Typos)**: We appreciate the reviewer’s careful review and have corrected the identified misprints and typos in the revised draft.
>
> **Question 1 (Citation of [Peyre & Cuturi])**: We apologize for the incorrect citation of this work, which is not relevant to the context of our paper. We have removed this reference and updated the draft accordingly.
>
> **Question 2 (Choice of $a_t^\theta$)**: Although not explicitly stated in the draft, it is possible to superpose multiple $a_t^\theta$ values, each corresponding to a different permutation of the coordinates.
>
> **Question 3 (Direction of the Flux)**: The flux expression $j_t$ is given in Equation (7) of the draft. If no additional terms $b_t$ are introduced, we have $j_t = \partial_t a_t$. Since $a_t^\theta$ has nonzero components across all coordinates, $j_t$ will be nonzero in all directions, and it is not necessarily parallel to the basis vectors. We also clarify that we provide the definition for the **i-th coordinate of $a_t^\theta$** rather than the entire vector.
>
> **Question 4 (Indices and Notations)**: We thank the reviewer for pointing out ambiguities regarding the indices and notations. We have clarified these in the revised draft.
>
> **Question 5 (Required Conditions for the Neural Network)**: We appreciate the reviewer’s correction. The correct condition is that $\sigma(x_D) \rightarrow 1$ as $x \rightarrow \infty$. This update has been incorporated into the revised draft.
>
> **Question 6 (Comparison to Baseline Methods)**: We do not believe a comparison to jump CNFs or attentive CNFs is appropriate, as these methods do not learn the marginal densities $\rho_t(x)$ as we do. Their conditioning on the event history enables them to achieve better NLLs, but this advantage arises from the non-Markovian nature of the time series they model, which differs from the problem we are addressing.
>
> **Question 7 (Kinetic Optimality)**: The additional divergence-free term $b_t$ introduced in Section 3.7 does not alter the density expression $\rho_t$, which depends solely on $a_t$. However, it does influence the flux and, consequently, the vector field $v_t$. The purpose of adding this term is to facilitate the training of kinetically optimal paths.
>
> **Question 8 (Figure 2)**: The "explosion" observed in the first column of Figure 2 occurs because our loss objective includes an entropy term that encourages the particles to spread out and explore different paths. In regions where no obstacles are present, the particle dynamics are primarily driven by this entropy term, leading to the observed behavior.
>
> **Comment (Comparison to Neural Conservation Laws)**: We make a comparison of our method and neural conservation laws in Section 2 and we would like to further elaborate our argument here.
>
> Neural conservation laws cannot be trained with maximum likelihood as the construction does not guarantee that the density integrates to 1 over the entire free domain. But we bake that constraint into the construction of our model. Our construction can be considered as a simplification and extension of neural conservation laws where we derive the core building blocks of neural conservation laws that are necessary for our approach following only basic principles and we remove the reliance on differential forms required by neural conservation laws.
>
> **Comment (Notation of $f_t^\theta$)**: We thank the reviewer for correcting this abuse of notations. We updated accordingly in the revised draft.

---

> > ### Comment · Reviewer_kXdF · 2024-11-29
> > **Thank you**
> >
> > I thank the authors for the provided response.
> >
> > At first, just a general comment (it does not affect my grade) - in the future, when updating the revision please mark the newly added text with, e.g., other color. It helps a lot tracking what was changed in the text.
> >
> > Some more specific comments:
> >
> > 1. They excluded [Bilos et. al.] from the related works (it is still presented in experiments, but it is strange that this closely related method is not mentioned in the related works section in the revised manuscript).
> >
> > 2. “Since $a_t^{\theta}$ has nonzero components across all coordinates”. Maybe I am missing something, but according to the eq. (12) $a_t^{\theta}$ has a nonzero component only across the $D$-th coordinate, so my question remains.
> >
> > 3. In my questions, I mentioned that Lemma 2 holds only for $i=D$. Seems that the authors agree with my opinion, but the formulation (and the proof) of the Lemma 2 remains the same.
> >
> > 4. Eqs. 21, 22, 23 - still there are a lot of places where $x_i$ should be substituted with $x_D$.
> >
> > 5. “We do not believe a comparison to jump CNFs or attentive CNFs is appropriate, as these methods do not learn the marginal densities” - I do not completely understand this point, since NLL metric is based on likelihood which is also modelled by Jump/Attentive CNF.
> >
> > Based on the comments above I vote for the rejection of the paper.
> >
> > Marin Biloš, Johanna Sommer, Syama Sundar Rangapuram, Tim Januschowski, and Stephan Gün-
> > nemann. Neural flows: Efficient alternative to neural odes. Advances in neural information
> > processing systems, 34:21325–21337, 2021.

---

### Author Response · Authors · 2024-11-27

We thank the reviewers for their careful reading of our paper and for ackowledging the contributions of our method. We reply to each reviewer separately below but we first address a common concern in this general response:

**Scalability**: Most reviewers are concerned about the scalability of our method and they are correct that our method is only as scalable as autoregressive (AR) models, due to its reliance on AR models for scaling to high-dimensional data. However, note that AR models are already used at scale for text, e.g. chatGPT, with significant engineering efforts such as KV caching significantly speeding up AR models. We unfortunately do not have the same expertise to enable fast AR models and used only small MLPs for our experiments. As such, we focused mainly on low-dimensional problem settings. Despite this, we have shown that there is significant promise in using our approach, outperforming all existing methods on the data sets that we considered. This signifies that it is worth spending significant effort in scaling up this approach with a fast AR model implementation for continuous space, which is a research question of its own. We note, however, that our proposed contributions are orthogonal to the choice of AR model.

---

### Meta-Review · Area_Chair_TrHe · 2024-12-20

**Metareview:**

This work proposes a simulation-free method for training diffusion processes over general objective functions. Inspired by Neural Conservation Laws, the authors develop a direct parameterization of a family of diffusion processes which are guaranteed to satisfy Fokker-Planck constraints. Reviewers agreed that the work was novel and generally well-written, with particular praise toward the theoretical developments. However, most also took issue with the experiments, most notably the lack of demonstrations for scalability into higher dimensions. This is troubling, as the corresponding simulation-based adjoint approaches became popular precisely due to their scalability. There are many ways that one can demonstrate scalability without involving an extremely expensive experiment, for example, wall times/memory usage can be reported for setups of increasing size. The new Appendix D shows some of the advantages over simulation approaches, but greater emphasis on these points seem necessary for adoption. I would strongly recommend that the reviewers resubmit to a future venue and consider including a grounded (possibly synthetic) experimental setup covering scalability and providing better reporting of wall times and memory usage.

**Additional Comments On Reviewer Discussion:**

Reviewers kXdF, hL5b, and k3XS all addressed the issue of scalability to higher dimensions. In response, the authors acknowledged this as a limitation, mentioning that small-scale experiments were conducted due to computational limitations. Reviewer k3XS acknowledged this response, but did not change their assessment. Reviewer hL5b pointed out the unusual lack of wall time reporting, which was partially addressed by the authors. Reviewer hL5b remained dissatisfied that some baselines were missing, as only those reported elsewhere were included. Reviewer kXdF commented on the lack of source code for the submission, and in response, the authors acknowledged that they would release the source code. Reviewer kXdF raised a few new issues in the revision, as well as some concerns regarding presentation of findings surrounding Lemma 2, which were not entirely addressed.

---

### Decision · Program_Chairs · 2025-01-22

Reject